# Predicting Hypnotic Use among Insomnia Patients with the Theory of Planned Behavior and Craving

**DOI:** 10.3390/bs12070209

**Published:** 2022-06-24

**Authors:** Chien-Ming Yang, Yu-Shuan Lai, Yun-Hsin Huang, Ya-Chuan Huang, Hsin-Chien Lee

**Affiliations:** 1Department of Psychology, National Cheng-Chi University, Taipei 116, Taiwan; yushuanlai@gmail.com (Y.-S.L.); veronicahuang519@gmail.com (Y.-H.H.); yachuangh@gmail.com (Y.-C.H.); 2The Research Center for Mind, Brain, and Learning, National Cheng-Chi University, Taipei 116, Taiwan; 3Department of Psychiatry, School of Medicine, College of Medicine, Taipei Medical University, Taipei 110, Taiwan; 4Department of Psychiatry & Sleep Center, Taipei Medical University Hospital, Taipei 110, Taiwan

**Keywords:** insomnia, hypnotics, theory of planned behavior, craving

## Abstract

While long-term hypnotic use is very common in clinical practice, the associated factors have been understudied. This study aims to explore the cognitive factors that might influence the long-term use of hypnotics based on the theory of planned behavior (TPB), and examines the moderating effect of craving between cognitive intention and actual hypnotic-use behavior at follow-up. A total of 139 insomnia patients completed a self-constructed TPB questionnaire to measure their attitude, subjective norm, perceived behavioral control, and behavioral intention of hypnotic use, as well as the Hypnotic-Use Urge Scale (HUS) to measure their craving for hypnotics. They were then contacted through phone approximately three months later to assess their hypnotic use. Hierarchical regression showed that perceived behavioral control was the most significant determinant for behavioral intention of hypnotic use. Behavioral intention, in turn, can predict the frequency of hypnotic use after three months. However, this association was moderated by hypnotic craving. The association was lower among the participants with higher cravings for hypnotic use. The findings suggest that the patients’ beliefs about their control over sleep and daily life situations, and their craving for hypnotics should be taken into consideration in the management of hypnotic use.

## 1. Introduction

Insomnia is a common health-related complaint that tends to be chronic in course. A longitudinal survey study in the United States found that 72.2% of the participants with symptoms of insomnia, at least three nights per week for at least one month at baseline, continued to report symptoms of insomnia after three years [1]. A recent Canadian longitudinal study showed that 86.0% of participants with symptoms of insomnia had the problem after one year, and 59.1% of them continued to experience insomnia after five years [2].

In light of the tendency of chronicity, most hypnotic medications are indicated for short-term use. The efficacy and safety of long-term hypnotic use remain an issue. Benzodiazepine (BZD) has been reported to be associated with adverse effects, such as daytime drowsiness, dizziness, light headache, and memory impairments [3,4,5,6,7,8,9,10,11], and is associated with tolerance, dependence, and withdrawal symptoms after long-term use [12,13,14,15]. Although non-benzodiazepine benzodiazepine receptor agonists (BzRAs) were claimed to be safer alternatives to BZDs, concerns about abuse, dependence, and withdrawal have increased over the past decade [16]. Hypnotic-induced complex sleep behaviors [17,18,19] and other side effects [20] have also been reported. Despite these concerns, it is still very common for patients with insomnia to use hypnotics for a prolonged period of time in clinical settings. Previous survey studies have reported that about two-thirds of the regular hypnotic users had used the drug for over a year, and about one-third for more than five years [21,22]. Another study in the United States also showed a significant increasing trend of hypnotic use from 1999 to 2014 in medium- and long-term use (at least >6 months), but not in short-term use [23].

Although long-term use of hypnotics is a common practice in clinical settings, studies addressing this issue are limited. Previous studies mostly focused on the symptomatology, demographic features of the users, hypnotic use history, and the characteristics of the medication that are associated with long-term hypnotic use [24,25,26]. Studies exploring the psychological factors that may contribute to long-term hypnotic use are limited. A recent qualitative study used framework analysis to explore the treatment choice process in patients with insomnia [27]. Three major themes were identified in the decision-making process of taking medication, which were the patients’ beliefs about their insomnia, treatment, and meaning of good sleep, the patients’ self-imposed boundaries in their treatment, and the third was the patients’ uptake of the treatments. The study concluded that the decision-making process of treatment choice in patients with insomnia is heavily grounded in their illness and treatment beliefs, prior treatment experiences, and external social factors. Thus, it is important to explore the influences of these psychological factors more systematically to assist patients in deciding, managing, and discontinuing their hypnotic use when needed.

Among the psychological models for health-related behaviors, the theory of planned behavior (TPB) has been used to portray the cognitive factors for the use and dependence of various substances, such as smoking, alcohol, marijuana, and psychoactive drugs [28,29,30,31]. According to the TPB, the intention of a behavior is influenced by three aspects of cognitive factors: attitude toward behavior, subjective norms, and perceived behavioral control. In turn, the intention can predict actual behavior. The favorable or unfavorable attitude toward behavior is the result of beliefs about the possible outcome and evaluation of such an outcome. Beliefs about the expectations from significant others and subjective importance of such expectations produce the subjective norms. Perceived behavioral control is the aggregate consequence of beliefs about the presence of factors that may facilitate or impede the execution of the behavior, as well as the perceived power of these factors. It can also directly predict actual behavior (see Figure 1) [32,33].

The three factors of TPB may have different predictive powers for the use of different substances [34]. The findings on TPB can be applied to design intervention programs and preventive strategies for substance use [35,36]. For example, attitudes toward BZD use, mainly perceived health benefits, had the strongest influence on the length of BZD use through the mediation of intention. In addition, experienced users were more likely to use BZDs when they had less control over drug taking, and inexperienced users were more influenced by the perceived attitude of the prescriber toward the use of BZDs [37,38].

Although previous studies have established TPB models for the use of various substances, the use of hypnotics could be very different due to the following reasons. First, the purpose for hypnotic use is to resolve a health-related problem (i.e., sleeplessness), which is different from the use of substances for pleasure or recreational purposes. Second, the timing for the use of hypnotics is specific, that is, during the night prior to bedtime, while most of the other substances can be used at different times of the day. Therefore, the factors contributing to the continuous use of hypnotics may be different from the use of other substances. Thus, the present study aimed to address the cognitive factors associated with chronic hypnotic use based on the TPB.

Another psychological construct that has been shown to have a great impact on substance use behaviors is craving. In the context of drug use, craving has been defined as an intense or compelling urge or desire to use drugs [39], and has been found to predict relapse after abstinence [40,41,42]. Although the use of hypnotics is supposed to be a medical decision making based on rational reasoning, it is not uncommon for hypnotic users to have difficulty in discontinuing hypnotic use. A recent study has shown attentional bias toward hypnotic-related stimuli with event-related potentials in long-term hypnotic users [43], suggesting the alternation of neurocognitive processes toward hypnotic-related cues. It is possible that some motivational and/or emotional factors, such as craving, may play a moderating role in the association between intention and actual drug use behavior. Thus, in addition to the cognitive factors addressed by the TPB, the present study further explored the role of craving in predicting continued use of hypnotic medications.

The present study examines the predictiveness of the TPB and craving on hypnotic use at follow-up with a prospective longitudinal design. The participants filled out self-reported questionnaires assessing the TPB factors and craving at baseline, and hypnotic use frequency was assessed by phone approximately three months later. It was hypothesized that the three cognitive factors of the TPB, namely, attitude, subjective norm, and perceived behavior control, can predict the intention of hypnotic use; the intention can further predict actual drug use behavior after three months. In other words, more favorable attitudes, more social pressures, and control factors regarding hypnotic use predict a higher intention to use hypnotics. Furthermore, craving was hypothesized to be a moderator in the association between intention and actual behavior of hypnotic use. Intention could better predict hypnotic use at follow-up in individuals with low cravings than in those with high cravings.

## 2. Materials and Methods

### 2.1. Participants

Insomnia patients who have taken hypnotics were recruited from the outpatient clinic of the Taipei Medical University Hospital. The inclusion criteria were as follows: (1) aged between 20 and 65 years; (2) meeting the criteria for insomnia disorder in the Diagnostic and Statistical Manual of Mental Disorders, Fifth Edition (DSM-5) [44]; (3) no current or history of major psychiatric disorders; (4) no current or history of major medical conditions; and (5) were prescribed hypnotic medications by a physician. A total of 161 patients with insomnia who met the inclusion criteria were recruited initially; 22 of them were excluded because of missing data. Thus, 139 (female = 98/70.5%; male = 41/29.5%) participants completed the three-month follow-up questionnaire. The mean age of participants was 45.91 (±11.69) years old, including 16 participants of 20–29 years old, 29 participants of 30–39 years old, 34 participants of 40–49 years old, 41 participants of 50–59 years old, and 19 participants of 60–65 years old. The duration of their insomnia is between 3 to 480 months, with 95.58 (±86.58) months on average. The duration of their hypnotic use was between 1 and 240 months, with an average of 58.23 (±59.29) months. The types of hypnotic medication used are listed in Table 1.

### 2.2. Measurements

#### 2.2.1. Theory of Planned Behavior Questionnaire

The self-developed TPB questionnaire was constructed following the standard procedure described by Ajzen [45]. Twenty-one patients with insomnia were initially interviewed to collect their beliefs about the possible consequences of hypnotic use, the expectations of others on hypnotic use, and the presence of situational factors that may facilitate or impede hypnotic use; the answers from the patients were integrated and sorted according to the number of times reported to serve as the basis for the subscales of attitude, subjective norm, and perceived behavioral control of the TPB questionnaire, respectively. Similar responses reported by more than 80% of the patients were retained as items of the scales, indicating good content validity. Following the manual for TPB questionnaire construction, we used indirect measurements to assess attitudes, subjective norms, and perceived behavioral control, and the generalized form to assess behavioral intention [46]. All items were assessed on a seven-point Likert form. On positive and negative attitude subscales, items of behavioral beliefs (e.g., taking hypnotics makes me fall asleep quickly; scored 1 to 7) and corresponding items of outcome evaluations (e.g., how much I want to “fall asleep quickly”; scored −3 to 3) were multiplied. The scores of each multiplied pair were summed for the total score of positive attitudes (PA, 7 items, total range from −147 to 147) and negative attitudes (NA, 4 items, total range from −84 to 84). Higher scores indicated stronger positive or negative attitudes toward hypnotic use. On the subjective norm (SN) subscale, the normative beliefs of physicians, family, and friends (scored −3 to 3), and corresponding motivation to comply (scored 1 to 7) were multiplied, resulting in a total score range of −63 to 63. Higher scores indicated stronger social pressure to use hypnotics. Perceived behavioral control was multiplied by the strength of control beliefs (e.g., feeling aroused around bedtime; scored 1 to 7) and the power of these factors (e.g., when I feel aroused around bedtime, I would take hypnotics; scored −3 to 3). The summed score of 18 items of perceived behavioral control ranged from −378 to 378; higher scores indicated stronger facilitating factors of hypnotic use. The behavioral intention (BI) subscale included three items (scored 1 to 7), with a total range from 3 to 21; higher scores indicated stronger intention to use hypnotics. The Cronbach’s alpha of all subscales ranged from 0.74 to 0.84, indicating good internal consistency.

#### 2.2.2. Hypnotic-Use Urge Scale (HUS)

The HUS assesses urge and craving for hypnotics at bedtime. This 37-item self-reporting questionnaire contains three subscales: (1) anticipated effects (e.g., “Taking sleeping pills before going to bed makes me feel relaxed”); (2) compelling desire to use hypnotics (e.g., “I must take sleeping pills before going to bed”); (3) preoccupation and pleasurable feelings (e.g., “At bedtime, I feel the sleeping pills tempting me.”). Participants were required to rate their level of agreement on each item on a Likert-type scale (1 = strongly disagree, 7 = strongly agree). The HUS has good internal consistency, test–retest reliability, concurrent validity, and construct validity [47].

### 2.3. Procedure

The potential participants went through an interview for the inclusion criteria first by a trained researcher. Psychiatric disorders were screened using the Mini International Neuropsychiatric Interview [48], and other sleep disorders were screened using the Sleep-50 [49]. Participants who passed the screening were then required to fill in a package of questionnaires, including a survey for demographic data, a TPB questionnaire to assess the cognitive factors associated with hypnotic use, and the HUS to assess the craving for hypnotics. Informed consent was obtained before administering the questionnaires. The participants were then contacted by phone approximately three months later for follow-up assessments of their hypnotic use, including the average frequency of hypnotic use in a week in the past month, and the type and dosage of hypnotics used.

### 2.4. Statistical Analysis

Multiple regression and hierarchical multiple regression were conducted to test the mediation effect according to the procedure of Baron and Kenny [50]. In step 1, independent variables (attitudes, subjective norm, perceived behavioral control) were entered to predict the mediator (behavioral intention). In step 2, the mediator was used to predict targeted behavior (hypnotic use). In step 3, all the independent variables were entered in the first hierarchy, and the mediator was entered in the second hierarchy to test the effect of the mediator on targeted behavior when independent variables were controlled. In addition, the Sobel test [51] was done to further validate the mediation effect. The moderation effect of craving was tested by entering the interaction effect as a predictor by multiplying the mean-centered scores of HUS subscales and behavioral intention. The attitudes, subjective norm, perceived behavioral control, behavior intention, and HUS subscale were entered in the first hierarchy, and the interaction effect was entered in the second hierarchy.

## 3. Results

### 3.1. The TPB Model

The TPB model was examined using hierarchical regressions, and the relevant coefficients are listed in Table 2. All the variance inflation factor coefficients were between 1.03 and 3.37, indicating that there was no significant problem regarding collinearity. Among the three variables of the TPB, only the perceived behavioral control (*β* = 0.41, *p* < 0.001) showed significant predictive power for behavioral intention. This can explain 23.6% of the variability in behavioral intention. Behavioral intention (*β* = 0.54, *p* < 0.001) was also shown to predict the frequency of hypnotic use at follow-up. Perceived behavioral control was also shown to have significant predictive power for hypnotic use at the three-month follow-up after controlling for the contribution of behavioral intention (*β* = 0.27, *p* < 0.01). Thus, behavioral intention serves as a partial mediator of the association between perceived behavioral control and actual drug use behavior. The Sobel test supported the mediation effect (Z = 3.65, *p* < 0.001). The total model, including perceived behavioral control and behavioral intention, can explain 34.2% of the variance in hypnotic use behavior at follow-up. A visual illustration is presented in Figure 1A.

### 3.2. Moderation Effect of Craving

Regarding the moderation effect of craving on the association between behavioral intention and the frequency of hypnotic use, the regression model showed no significant moderating effect of the total HUS score. However, when using the subscale scores as moderators, the subscales for compelling desire (BI x CD, *β* = −0.13, *p* < 0.05), and preoccupation and pleasurable feelings (BI x PP, *β* = −0.17, *p* < 0.05) showed significant moderation effects, while the subscale for anticipated effects showed no significant moderation effect (Table 3). The higher the scores on the first two subscales, the lesser the degree to which behavioral intention can predict actual drug use behavior. A visual illustration is presented in Figure 1B,C.

## 4. Discussion and Conclusions

The current study explored the role of cognitive beliefs and intention in predicting the use of hypnotic medication and the moderating effect of an emotional factor (i.e., craving) on the predicting effect. Similar to the findings of previous studies on substance use behaviors based on the TPB model [30,31], the behavioral intention determined by cognitive beliefs could significantly predict the frequency of hypnotic use three months later, and could explain 34.2% of the variance of the behavior. This suggests that the use of hypnotic medication is in part based on the cognitive process of decision making.

Nonetheless, among the cognitive factors that determine behavioral intention, perceived behavioral control was found to be the most significant one. This might reflect that the intention to use hypnotics is highly relevant to the need for control over situational factors. When all three factors were included in a regression model to predict hypnotic use intention, perceived behavioral control was the only factor that remained significant. The other two components, attitude and subjective norm, had relatively less influence on the intention to continue or discontinue hypnotic use. This is consistent with the finding in a meta-analysis that perceived behavioral control to have the largest association with the intention of chronic illness control behaviors compared with the other two cognitive constructs in the TPB model [52]. Perceived behavioral control refers to an individual’s belief in the ease or difficulty to perform the behavior of interest. This is, in turn, determined by the individual’s belief about the occurrences of situational factors that may facilitate or impede the performance of the behavior. In the context of hypnotic use, some of the facilitating factors are sleep-related, such as “having trouble sleeping for consecutive nights” and “feeling aroused around bedtime”; others are more related to daytime situations, such as “having important thing the next day” or “experiencing emotional disturbance in the day.” The impeding factors, on the other hand, are more of a general feeling or are associated with their daytime conditions, such as “feeling emotionally stable”, “too busy during the day”, or “being off duty the next day”. Thus, patients’ intention of hypnotic use is more determined by their beliefs about their control over situational factors than the beliefs about the benefits and risks of hypnotic use. The beliefs about behavioral control are closely related to the concept of self-efficacy, which has also been shown to predict drug discontinuation in previous studies [53,54].

It is still a surprise that the attitude about hypnotic use neither of oneself nor significant others (i.e., family, friends, and physicians) did not contribute much to the intention to use hypnotics. This is probably because hypnotic use is perceived as the only medical treatment for insomnia. Without knowing the other alternatives, such as cognitive-behavioral therapy for insomnia, and with sleep being a necessary part of life, the beliefs about the benefits and risks of drug use might not have much influence on their decision to use hypnotics. Moreover, the use of hypnotics might be in part a strategy to deal with daily life stressors rather than to treat insomnia per se, as has been reported in a qualitative study of long-term hypnotic users [55]. Thus, what determines patients’ intention of hypnotic use is more the daily life situations that are beliefs to be associated with their sleep difficulties. This may also resemble the drug use pattern of “as-needed use”, which is commonly seen in hypnotic users. The findings also speak to the importance of public education about the options for the treatment of insomnia.

Another important finding of the present study is that craving serves as a moderator between the intention and the actual behavior of drug use. The intention to use hypnotics could predict hypnotic use after three months only in participants with lower levels of craving, but not in those with higher levels of craving. It is worth noting that particularly the subscales of “compelling desire to use hypnotics” and “preoccupation and pleasurable feelings from hypnotic use” showed significant moderation effects. These two subscales reflect the types of craving commonly seen in the use of addictive substances. Craving has long been an independent predictor of relapse or a mediator between drug cues and addictive behaviors [56,57,58,59]. This study further suggests that high craving may interfere with an individual’s decision making around substance use behavior, and lead to a more “uncontrollable” nature of substance use.

In summary, the findings of the current study suggest that the decision to use hypnotics is less influenced by patients’ beliefs about the overall pros and cons of hypnotic use, but more related to the situational factors associated with their mood, physical conditions, and stress at bedtime or during the day. Furthermore, in patients who have developed strong cravings, their intension based on cognitive evaluation does not really predict actual drug use behavior few months later. This indicates that the compelling drug “wanting”, as proposed by Robinson and Berridge [60], in the drug addiction process, might have been developed in these patients. These findings have important implications, particularly for the discontinuation of long-term hypnotic use. First, since the intention of hypnotic use is more associated with daily life situations, it is important to help patients deal with not only their sleep problems but also their daytime issues when they are ready to taper off the medication. Adding stress management to CBT-I may be helpful. Second, some long-term hypnotic users may have developed addiction to the medication. It is important for the physician to evaluate the level of craving in patients who are about to discontinue hypnotic use. It could be essential to provide strategies to manage their craving and compelling urge of drug use in order to discontinue hypnotic use successfully. Furthermore, through the clarification of patients’ perceived behavioral control, physicians could possibly enhance patients’ sense of control to reduce hypnotic use with psychoeducation.

In light of these implications, the limitations of the study should also be taken into consideration. First, the follow-up period was three months. It is not certain whether the findings will be consistent at a longer follow-up time. Second, the sample size of the current study was small, as some of the participants dropped out during the follow-up period. Third, our study did not assess the patients’ response to hypnotics and did not track the changes in hypnotic prescription. Therefore, we could not address some important issues such as therapy-resistant insomnia or inadequate treatment. The effect of these factors on continuous hypnotic use should be further investigated in future study.

## Figures and Tables

**Figure 1 behavsci-12-00209-f001:**
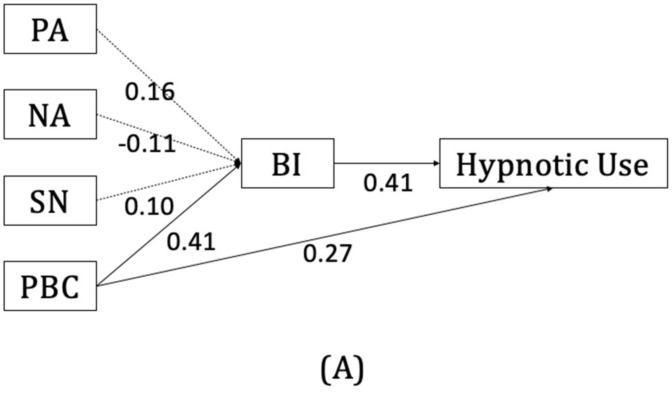
The visual illustration of coefficients of the Theory of Planned Behavior (TPB) constructs and moderation effect of craving on hypnotic use: (**A**) The predicting values of TPB constructs on hypnotic use; (**B**) The moderation effect of Compelling Desire (CD) on the association between behavioral intention and hypnotic use; (**C**) The moderation effect of Preoccupation and Pleasurable feelings (PP) on the association between behavioral intention and hypnotic use. (PA: positive attitude; NA: negative attitude; SN: subjective norm; PBC: perceived behavioral control; BI: behavioral intention).

**Table 1 behavsci-12-00209-t001:** The number and percentage of participants using different hypnotic medications.

Medication	N (%)	Medication	N (%)
BZD		BzRAs	
Estazolam	36 (21.3%)	Zolpidem	45 (26.6%)
Clonazepam	23 (13.6%)	Zopiclone	6 (3.6%)
Lorazepam	15 (8.9%)		
Alprazolam	12 (7.1%)	Antidepressants/Antipsychotics	
Bromazepam	7 (4.1%)	Trazodone	7 (4.1%)
Flunitrazepam	5 (3.0%)	Mirtazapine	2 (1.2%)
Diazepam	3 (1.8%)	Sulpiride	3 (1.8%)
Brotizolam	2 (1.2%)		
Triazolam	2 (1.2%)		
Fludiazepam	1 (0.6%)		

BZD: benzodiazepine; BzRAs: non-benzodiazepine benzodiazepine receptor agonists.

**Table 2 behavsci-12-00209-t002:** Hierarchical regression coefficients for the test of the Theory of Planned Behavior to predict medication use at the 3-month follow-up.

Predictor	*B*	*β*	*t*
**Step 1**	**DV: Behavioral Intention**		
Attitude (positive)	0.01	0.16	1.93
Attitude (negative)	−0.01	−0.11	−1.42
Subjective Norm	0.02	0.10	1.23
Perceived Behavioral Control	0.02	0.41	5.02 ***
*R*^2^ = 0.26; *Adj. R*^2^ = 0.24 *F*(4, 134) = 11.64; *p* < 0.001			
**Step 2**	**DV: Hypnotic Use**		
Behavioral Intention	1.23	0.54	7.49 ***
**Step 3**	**DV: Hypnotic Use**		
Model 1			
Attitude (positive)	0.01	0.07	0.89
Attitude (negative)	0.01	0.04	0.48
Subjective Norm	0.03	0.07	0.93
Perceived Behavioral Control	0.05	0.43	5.32 ***
Model 2			
Attitude (positive)	0.00	0.01	0.12
Attitude (negative)	0.03	0.08	1.14
Subjective Norm	0.01	0.04	0.48
Perceived Behavioral Control	0.03	0.27	3.30 **
Behavioral Intention	0.95	0.41	5.07 ***
*R*^2^ = 0.37; *Adj. R*^2^ = 0.34 *F*(5, 132) = 15.36; *p* < 0.001			

** *p* < 0.01; *** *p* < 0.001.

**Table 3 behavsci-12-00209-t003:** The moderation effects of the subscales of the Hypnotic-use Urge Scale on the association between behavioral intention and frequency of hypnotic use.

Subscale 1: Anticipated Effects					Subscale 2: Compelling Desire					Subscale 3: Preoccupation and Pleasurable Feelings				
Predictor	*B*	*β*	*t*		Predictor	*B*	*β*	*t*		Predictor	*B*	*β*	*t*	
**Model 1**					**Model 1**					**Model 1**				
PA	−0.03	−0.18	−1.75		PA	−0.02	−0.12	−1.79		PA	−0.02	−0.12	−1.38	
NA	0.03	0.08	1.20		NA	0.01	0.04	0.73		NA	0.03	0.10	1.42	
SN	0.01	0.03	0.37		SN	0.02	0.06	0.94		SN	0.01	0.02	0.25	
PBC	0.02	0.18	2.15 *		PBC	0.00	0.04	0.52		PBC	0.02	0.22	2.76 **	
BI	0.81	0.35	4.31 ***		BI	0.35	0.15	1.97		BI	0.81	0.35	4.33 ***	
AE	0.27	0.32	2.73 **		CD	0.60	0.64	7.19 ***		PP	0.48	0.28	3.17 **	
**Model 2**					**Model 2**					**Model 2**				
PA	−0.02	−0.13	−1.30		PA	−0.02	−0.09	1.38		PA	−0.02	−0.09	−1.10	
NA	0.03	0.08	1.22		NA	0.02	0.05	0.84		NA	0.02	0.08	1.15	
SN	0.01	0.03	0.47		SN	0.02	0.05	0.81		SN	0.01	0.02	0.26	
PBC	0.02	0.18	2.08 *		PBC	0.00	0.03	0.38		PBC	0.02	0.22	2.74 **	
BI	0.86	0.37	4.52 ***		BI	0.41	0.18	2.30 *		BI	0.82	0.35	4.49 ***	
AE	0.21	0.25	2.05 *		CD	0.57	0.60	6.68 ***		PP	0.48	0.28	3.22 **	
BIxAE	−0.02	−0.12	−1.60		BIxCD	−0.02	−0.13	−2.05 *		BIxPP	−0.05	−0.17	−2.54 *	
**Model Comparison**														
	*R* ^2^	*Adj. R* ^2^	Δ*R*^2^	Δ*F*		*R* ^2^	*Adj. R* ^2^	Δ*R*^2^	Δ*F*		*R* ^2^	*Adj. R* ^2^	Δ*R*^2^	Δ*F*
**Model 1**	0.40	0.37	0.40	14.66 ***		0.55	0.52	0.55	26.31 ***		0.41	0.38	0.41	15.35 ***
**Model 2**	0.41	0.38	0.01	2.55		0.56	0.54	0.01	4.21 *		0.44	0.41	0.03	6.46 *

PA = Positive Attitude; NA = Negative Attitude; SN = Subjective Norm; PBC = Perceived Behavioral Control; BI = Behavioral Intention; AE = Anticipated Effects; CD = Compelling Desire; PP = Preoccupation and Pleasurable feelings. * *p* < 0.05; ** *p* < 0.01; *** *p* < 0.001.

## Data Availability

The data presented in this study are available on request from the corresponding author. The data are not publicly available due to the fact that we did not obtain the participants’ consent to have the data available for the public.

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
