# Peer review of "Predicting Hypnotic Use among Insomnia Patients with the Theory of Planned Behavior and Craving"

_behavsci, 2022, doi:10.3390/bs12070209_

Round 1
Reviewer 1 Report
This is an important study that investigated the predicting factors for hypnotic use among insomnia patients using the theory of planned behavior. The manuscript is well written, however, the following points need to be clarified before acceptance for publication:
- Was there any difference (statistically) on the prevalence of insomnia between male and female participants? why you did not report on this?
- How many participants were in different age groups? e.g 20-30 years of age?
- Line 256-259: references are missing. Previous studies with similar findings should be cited.
- Line 262- Why the perceived behavioral control was significantly different compared to other variables?
I applaud the authors for ideal efforts they put when preparing the manuscript. It is scientifically sound and the methodology is very detailed.
Theory
The manuscript is well written and informative. Authors succeeded in providing a detailed and acceptable rationale describing why their study, compared to other contemporary studies, is novel and adding new knowledge to the field of behavioral sciences. The knowledge gap is also addressed and problem statement is well justified.
Methodology
Although I pointed out few issues that needs clarity. i.e. How many participants were in different age groups? This must be discussed, Why the prevalence of insomnia was not compared between male and female participants?, the methodology is well described to address the study aim. Also, the choice of statistical analysis was carefully considered to address the study objective, and I applaud the authors for that.
Conclusion
The conclusion transmit the main ideas clearly and reflect what the readers as a take-home message. Concluding findings and implications are clearly stated.
There are missing references under the discussion; line 256-259. Previous studies with similar findings, as stated, should be cited. And also, why the perceived behavioral control was significant compared to other variables- line 262.
Author Response
This is an important study that investigated the predicting factors for hypnotic use among insomnia patients using the theory of planned behavior. The manuscript is well written, however, the following points need to be clarified before acceptance for publication:
- Was there any difference (statistically) on the prevalence of insomnia between male and female participants? why you did not report on this?
Response 1: Thank you for raising this issue. Our study was designed to investigate hypnotic use in insomnia patients. All our participants had insomnia, thus we do not have data on the prevalence of insomnia among general population. We did report the numbers of female and male participants in our study (see Page.3, line 128; female= 98/70.5%; male = 41/29.5%),
- How many participants were in different age groups? e.g 20-30 years of age?
Response 2: We have added the numbers of participants in different age groups (see Page 3, lines 129- Page 4,132) as the following:
“The mean age of participants was 45.91 (± 11.69) years old, comprised of following groups: 16 participants as 20-29, 29 participants as 30-39; 34 participants as 40-49; 41 participants as 50-59; and 19 participants as 60-65.”
- Line 256-259: references are missing. Previous studies with similar findings should be cited.
Response 3: We thank the reviewer for pointing out the missing citation. We have added the citations ([30,31]) in the text (See Page 8, line 251).
- Line 262- Why the perceived behavioral control was significantly different compared to other variables?
Response 4: Thank you for letting us to clarify this part. We had some discussion in the second (p.8) and the fourth (p.9) paragraphs of the Discussion and Conclusion section. The evidence from our results, in comparison with previous literature, and possible explanations were discussed. We also added a sentence, “This might reflect the intention to use hypnotics is highly relevant to the need to control over situational factors.” (see Page 8, lines 256-258).
I applaud the authors for ideal efforts they put when preparing the manuscript. It is scientifically sound and the methodology is very detailed.
Theory
The manuscript is well written and informative. Authors succeeded in providing a detailed and acceptable rationale describing why their study, compared to other contemporary studies, is novel and adding new knowledge to the field of behavioral sciences. The knowledge gap is also addressed and problem statement is well justified.
Methodology
Although I pointed out few issues that needs clarity. i.e. How many participants were in different age groups? This must be discussed, Why the prevalence of insomnia was not compared between male and female participants?, the methodology is well described to address the study aim. Also, the choice of statistical analysis was carefully considered to address the study objective, and I applaud the authors for that.
Conclusion
The conclusion transmit the main ideas clearly and reflect what the readers as a take-home message. Concluding findings and implications are clearly stated.
There are missing references under the discussion; line 256-259. Previous studies with similar findings, as stated, should be cited. And also, why the perceived behavioral control was significant compared to other variables- line 262.
Response: We thank the reviewer for the positive feedback and careful review to point out our problems. We have revised our manuscript according to the comments and replied to each comments above.
Reviewer 2 Report
In this manuscript Yang et al. used theory of planned behavior and craving for predicting use of hypnotics among patients with insomnia. The paper is important, as insomnia is a frequent sleep complain, which influence quality of life.
I some comments:
1.) page 3 - line 130 - please mention the name of the General hospital
2.) I am somehow concern regarding the criteria 4, page 3, line 133, ''no current or history of major medical conditions'', the recruited patients being addmitted to the hospital, so with some more serios medical condition
3.) it would be helpfull to know if the study was prospective or retrospectiv
4.) all references need to be checked carefully, as they are inconsistencies like page 4, line 147 Azjen [46], which is actually [47]
page 4, line 187-188, reference [49] is actually [50], reference [50] is actually [51]. The number of references may be reduced overall.
5.) the paper could be shortened, for instance introduction is too long, in my opinion and should be more focused.
Author Response
In this manuscript Yang et al. used theory of planned behavior and craving for predicting use of hypnotics among patients with insomnia. The paper is important, as insomnia is a frequent sleep complain, which influence quality of life.
I some comments:
- page 3 - line 130 - please mention the name of the General hospital
Response 1: We have added the name of the general hospital (“Taipei Medical University Hospital”) to the Participant section (Page 3, lines 121-122) .
2.) I am somehow concern regarding the criteria 4, page 3, line 133, ''no current or history of major medical conditions'', the recruited patients being addmitted to the hospital, so with some more serios medical condition
Response 2: We thank the reviewer for the concern. The patients were recruited from the outpatient clinic of the general hospital. We have revised the sentence to make it more clear (Page 3, lines 121-122). It is very common in Taiwan that patients attend outpatient clinic of a general hospital due to rather minor symptoms such as sleep problems.
3.) it would be helpfull to know if the study was prospective or retrospective
Response 3: This is a longitudinal study in which the participants filled out a package of questionnaires at the initial participation, and the medication use were re-assessed at follow-up by phone three months later. We added description of the study design to make it more clear (see Page 3, lines 107-110) as follow:
“The present study examines the predictiveness of the TPB and craving on hypnotic use at follow-up with a prospective longitudinal design. The participants filled out self-reported questionnaires assessing the TPB factors and craving at baseline, and hypnotic use frequency was assessed by phone approximately three months later.”
4.) all references need to be checked carefully, as they are inconsistencies like page 4, line 147 Azjen [46], which is actually [47] page 4, line 187-188, reference [49] is actually [50], reference [50] is actually [51]. The number of references may be reduced overall.
Response 4: We thank the reviewer to point out the problem in citation. We have rechecked all the references to make sure they are consistent with the citation in the text.
5.) the paper could be shortened, for instance introduction is too long, in my opinion and should be more focused.
Response 5: Thank you for the suggestion. We have revised the Introduction by combining some of the descriptions of different previous studies to make it slightly shorter (See Page 2, the 2nd and 4th paragraphs of the Introduction section).
Reviewer 3 Report
This manuscript deals with an important and interesting topic. The background is clearly described in the Introduction. Methods are timely and appropriate. The manuscript is well written.
However tthe following aspects should also in addition be taken into account.
1. To what extent did at least some of the patients suffer from therapy-resistant or inadequately treated insomnia that led to chronic use of hypnotics?
2. Was this measured by subjective or objective parameters?
3. Did all patients continuously receive the same hypnotic drug mentioned in table 1 or was the substance changed in some patients, e.g. due to a lack of therapeutic success?
4. The role of physicians in the intake of hypnotics is mentionned in the Introducution is, however, no longer taken into account in research questions, in the Results and in the Discussion.
Minor issues
line 25 - please add "of"
line 29 - please errase s in comnplaint
Table 1 - please correct to Trazodone
Author Response
This manuscript deals with an important and interesting topic. The background is clearly described in the Introduction. Methods are timely and appropriate. The manuscript is well written.
Response: Thank you for your affirmation.
However tthe following aspects should also in addition be taken into account.
- To what extent did at least some of the patients suffer from therapy-resistant or inadequately treated insomnia that led to chronic use of hypnotics?
Response 1: We thank the reviewer for raising these important issues. However, the data of hypnotic use in our study was collected through a survey questionnaire. We did not have detail information of hypnotic use and patients’ response due to the survey nature of the study. We therefore could not answer the questions regarding therapy-resistant or inadequately treated insomnia. We agree that these are important issues and have added it as a limitation of our study in the 4. Discussion and Conclusions (lines 325-328).
“Third, our study did not assess the patients’ response to hypnotics and did not track the changes in hypnotic prescription. Therefore, we could not address some important issues such as therapy-resistant insomnia or inadequate treatment. The effect of these factors on continuous hypnotic use should be further investigated in future study.”
- Was this measured by subjective or objective parameters?
Response 2: Our study is a survey study, all of our variables were assessed by self-report.
- Did all patients continuously receive the same hypnotic drug mentioned in table 1 or was the substance changed in some patients, e.g. due to a lack of therapeutic success?
Response 3: Thank you for raising this important question. As stated in our reply to the reviewer’s question 1, the current study did not track the changes in hypnotic prescription. We agree that this is an important issue and should be considered as a limitation of our study. We have added relevant discussion in line 325-328, as we stated in our reply to question 1.
- The role of physicians in the intake of hypnotics is mentionned in the Introducution is, however, no longer taken into account in research questions, in the Results and in the Discussion.
Response 4: We thank the reviewer for bringing up this important issue. The attitude of physician was incorporated in the factor of “subjective norm” under the model of Theory of Planned Behavior. It was however found not to be significant in predicting hypnotic use intention when perceived behavioral control was added into the model. We have revised our manuscript to make it clearer (Lines 278-280), with the following sentence:
“It is still a surprise that the neither the attitude about hypnotic use of oneself nor significant others ( i.e. family, friends, and physicians) did not contribute much to the intention to use hypnotics.”
We also suggested strategies for the physicians to help the patients discontinuing hypnotic use based on the findings (see Lines 314-320), with the following:
“It is important for the physician to evaluate the level of craving in patients who are about to discontinue hypnotic use. It could be essential to provide strategies to manage their craving and compelling urge of drug use in order to discontinue hypnotic use successfully. Furthermore, through the clarification of patients’ perceived behavioral control, physicians could possibly enhance the patients’ sense of control to reduce hypnotic use with psychoeducation.”
Minor issues
line 25 - please add "of"
Response: We thank reviewer for the careful examination. We have made the correction.
line 29 - please errase s in comnplaint
Response: Thank you for pointing out the mistake. We have corrected it.
Table 1 - please correct to Trazodone
Response: Thank you for pointing out our typo. We have corrected it.
Round 2
Reviewer 3 Report
The manuscript is in good shape. Authors improved it well in accordance to the reviewers' comments.
I ask for one minor correction. In line 277 please errase "the" in "that the neither"
Author Response
Dear Reviewer:
Thank you for pointing out the error in our revised manuscript. We have removed the extra "the" in line 277.
Thanks again for your thorough review and comments, and your positive feedback.